# Violence prevention accelerators for children and adolescents in South Africa: A path analysis using two pooled cohorts

**Lucie D. Cluver**[1,2‡¤*], **William E. Rudgard**[1‡], **Elona Toska**[3], **Siyanai Zhou**[4], **Laurence Campeau**[5], **Yulia Shenderovich**[1], **Mark Orkin**[1,6], **Chris Desmond**[7], **Alexander Butchart**[8], **Howard Taylor**[9], **Franziska Meinck**[10,11], **Lorraine Sherr**[12]

1 Department of Social Policy and Intervention, University of Oxford, Oxford, United Kingdom, 2 Department of Psychiatry and Mental Health, University of Cape Town, Cape Town, South Africa, 3 Centre for Social Science Research, University of Cape Town, Cape Town, South Africa, 4 Department of Statistics and AIDS and Society Research Unit, University of Cape Town, Cape Town, South Africa, 5 Oxford Research South Africa, East London, South Africa, 6 Medical Research Council Development Pathways to Health Research Unit, School of Clinical Medicine, University of the Witwatersrand, Johannesburg, South Africa, 7 Centre for Rural Health, University of KwaZulu-Natal, Durban, South Africa, 8 Violence Prevention Unit, Social Determinant of Health, Healthier Populations Division, World Health Organization, Switzerland, 9 Global Partnership to End Violence Against Children, New York, New York, United States of America, 10 OPTENTIA Faculty of Health Sciences, North-West University, South Africa, 11 School of Social and Political Science, University of Edinburgh, Edinburgh, United Kingdom, 12 Health Psychology Unit, Institute of Global Health, University College London, United Kingdom

¤ Current address: Department of Social Policy and Intervention, Barnett House, University of Oxford, Oxford, United Kingdom
‡ These authors contributed equally and are joint first authors on this work.
* lucie.cluver@spi.ox.ac.uk

**Data Availability Statement:** All relevant data are within the manuscript and its Supporting Information files.

## Abstract

### Background

The INSPIRE framework was developed by 10 global agencies as the first global package for preventing and responding to violence against children. The framework includes seven complementary strategies. Delivering all seven strategies is a challenge in resource-limited contexts. Consequently, governments are requesting additional evidence to inform which 'accelerator' provisions can simultaneously reduce multiple types of violence against children.

### Methods and findings

We pooled data from two prospective South African adolescent cohorts including Young Carers (2010–2012) and Mzantsi Wakho (2014–2017). The combined sample size was 5,034 adolescents. Each cohort measured six self-reported violence outcomes (sexual abuse, transactional sexual exploitation, physical abuse, emotional abuse, community violence victimisation, and youth lawbreaking) and seven self-reported INSPIRE-aligned protective factors (positive parenting, parental monitoring and supervision, food security at home, basic economic security at home, free schooling, free school meals, and abuse response services). Associations between hypothesised protective factors and violence

**Funding:** This study was funded by the UK Research and Innovation Global Challenges Research Fund (UKRI GCRF) Accelerate Hub [ES/S008101/1]; Claude Leon Foundation [F08 559/C]; The South African National Department of Social Development [27/2011/11 HIV AND AIDS]; The Health Economics and HIV/AIDS Research Division (HEARD) at the University of KwaZulu-Natal [R14304/AA002]; Nuffield Foundation [OPD/31598] and [R46194/AA001]; South African National Research Foundation [RES-062-23-2068]; the John Fell Fund [103/757] & [161/033]; University of Oxford's Economic and Social Research Council Impact Acceleration Account (IAA) [K1311-KEA-004 ] and [1602-KEA-189]; the International AIDS Society through a Collaborative Initiative for Paediatric HIV Education and Research (CIPHER) grant [155-Hod] and [2018/625-TOS]; Evidence for HIV Prevention in Southern Africa, a UKAID programme managed by Mott MacDonald; Janssen Pharmaceutica NV, part of the Janssen Pharmaceutical Companies of Johnson & Johnson; Oak Foundation [R46194/AA001] and [OFIL-20-057]; the Regional Inter-Agency Task Team for Children Affected by AIDS - Eastern and Southern Africa (RIATT-ESA); the Leverhulme Trust [PLP-2014-095]; UNFPA South Africa; UNICEF Eastern and Southern Africa Office; Research England [0005218]; UCL's HelpAge funding; The European Research Council (ERC) under the European Union's (EU) Seventh Framework Programme (FP7/2007-2013) [313421]; the European Research Council (ERC) under the European Union's Horizon 2020 research and innovation programme [771468]; Fogarty International Center, National Institute on Mental Health (NIMH), National Institute of Health (NIH) [K43TW011434]; the UK Medical Research Council (MRC) and the UK Department for International Development (DFID) under the MRC/DFID Concordat, and by the Department of Health Social Care (DHSC) through its National Institutes of Health Research (NIHR) [MR/R022372/1]; and the University of Oxford Clarendon-Green Templeton College Scholarship. The funders had no role in study design, data collection, and analysis, decision to publish, or preparation of the manuscript.

**Competing interests:** I have read the journal's policy and the authors of this manuscript have the following competing interests: CD reports personal fees from University of KwaZulu-Natal during the conduct of the study. ET reports funding from UK Medical Research Council (MRC) and the UK Department for International Development (DFID), the European Research Council (ERC), UKRI GCRF Accelerating Achievement for Africa's Adolescents (Accelerate) Hub, the International AIDS Society

outcomes were estimated jointly in a sex-stratified multivariate path model, controlling for baseline outcomes and socio-demographics and correcting for multiple-hypothesis testing using the Benjamini-Hochberg procedure. We calculated adjusted probability estimates conditional on the presence of no, one, or all protective factors significantly associated with reduced odds of at least three forms of violence in the path model. Adjusted risk differences (ARDs) and adjusted risk ratios (ARRs) with 95% confidence intervals (CIs) were also calculated. The sample mean age was 13.54 years, and 56.62% were female. There was 4% loss to follow-up. Positive parenting, parental monitoring and supervision, and food security at home were each associated with lower odds of three or more violence outcomes ($p <$ 0.05). For girls, the adjusted probability of violence outcomes was estimated to be lower if all three of these factors were present, as compared to none of them: sexual abuse, 5.38% and 1.64% (ARD: −3.74% points, 95% CI −5.31 to −2.16, $p <$ 0.001); transactional sexual exploitation, 10.07% and 4.84% (ARD: −5.23% points, 95% CI −7.26 to −3.20, $p <$ 0.001); physical abuse, 38.58% and 23.85% (ARD: −14.72% points, 95% CI −19.11 to −10.33, $p <$ 0.001); emotional abuse, 25.39% and 12.98% (ARD: −12.41% points, 95% CI −16.00 to −8.83, $p <$ 0.001); community violence victimisation, 36.25% and 28.37% (ARD: −7.87% points, 95% CI −11.98 to −3.76, $p <$ 0.001); and youth lawbreaking, 18.90% and 11.61% (ARD: −7.30% points, 95% CI −10.50 to −4.09, $p <$ 0.001). For boys, the adjusted probability of violence outcomes was also estimated to be lower if all three factors were present, as compared to none of them: sexual abuse, 2.39% to 1.80% (ARD: −0.59% points, 95% CI −2.24 to 1.05, $p$ = 0.482); transactional sexual exploitation, 6.97% to 4.55% (ARD: −2.42% points, 95% CI −4.77 to −0.08, $p$ = 0.043); physical abuse from 37.19% to 25.44% (ARD: −11.74% points, 95% CI −16.91 to −6.58, $p <$ 0.001); emotional abuse from 23.72% to 10.72% (ARD: −13.00% points, 95% CI −17.04 to −8.95, $p <$ 0.001); community violence victimisation from 41.28% to 35.41% (ARD: −5.87% points, 95% CI −10.98 to −0.75, $p$ = 0.025); and youth lawbreaking from 22.44% to 14.98% (ARD −7.46% points, 95% CI −11.57 to −3.35, $p <$ 0.001). Key limitations were risk of residual confounding and not having information on protective factors related to all seven INSPIRE strategies.

## Conclusion

In this cohort study, we found that positive and supervisory caregiving and food security at home are associated with reduced risk of multiple forms of violence against children. The presence of all three of these factors may be linked to greater risk reduction as compared to the presence of one or none of these factors. Policies promoting action on positive and supervisory caregiving and food security at home are likely to support further efficiencies in the delivery of INSPIRE.

## Author summary

### Why was this study done?

- A billion children are victims of violence each year.

through the CIPHER grant. The views expressed in written materials or publications do not necessarily reflect the official policies of the International AIDS society. Research reported in this publication was supported by the Fogarty International Center, National Institute on Mental Health, National Institutes of Health. The content is solely the responsibility of the authors and does not represent the official views of the National Institutes of Health. FM reports grants from Economic and Social Research Council, grants from GCRF/ESRC during the conduct of the study and has previously consulted for the World Health Organization. LDC reports grants from the Global Challenges Research Fund (GCRF), the European Research Council, the Gates Foundation Grand Challenges, the HEFCE-GCRF Support Fund, the Medical Research Council, DFID, National Institute for Health Research. AB reports that he is a lead author of the "INSPIRE: Seven strategies for ending violence against children" technical package, and co-chair of the multi-agency INSPIRE Implementation Working Group. WER, YS, HT, SZ, LC, MO, and LS have nothing to disclose.

**Abbreviations:** ARD, adjusted risk difference; ARR, adjusted risk ratio; CDC, United States Centers for Disease Control and Prevention; CI, confidence interval; NGO, nongovernmental organisation; PAHO, Pan American Health Organization; PEPFAR, President's Emergency Plan For AIDS Relief; SD, standard deviation; SDG, Sustainable Development Goal; STROBE, Strengthening the Reporting of Observational Studies in Epidemiology; UNICEF, United Nations International Children's Fund; UNODC, United Nations Office on Drugs and Crime; USAID, US Agency for International Development.

- Governments need solutions that impact across not one but multiple childhood violence targets simultaneously—'development accelerators'.

- The World Health Organization's INSPIRE package of seven strategies for ending violence against children is our best starting point to test for these.

## What did the researchers do and find?

- We pooled two South African cohorts including 5,034 10- to 19-year-olds.

- We tested seven INSPIRE-aligned protective factors against six violence types: sexual abuse, transactional sexual exploitation, physical abuse, emotional abuse, community violence, and youth lawbreaking.

- Positive parenting, parental monitoring and supervision, and food security were each associated with lower likelihood of three or more types of violence.

- Experiencing all three of these factors was associated with lower likelihood of six types of violence for girls and boys, with up to 50% reductions.

## What do these findings mean?

- Caution should be taken, as observational studies cannot demonstrate causality.

- In the context of the COVID-19 economic crisis, strategic approaches to preventing violence are needed.

- Effective interventions to address positive and supervisory parenting and food security, such as parenting support and economic support, might reduce multiple forms of violence against children.

- When combined, these 'development accelerators' might have wider and stronger effects on multiple forms of violence.

## Introduction

We are at a pivotal moment of opportunity and risk in preventing and responding to violence against children. A billion children globally experience emotional, physical, or sexual violence annually, with evidence of increased rates of violence associated with the COVID-19 epidemic [1]. Severe, long-term impacts include higher mortality, morbidity, school dropout, unemployment, neurological deficits, sexual risks, HIV infection, adolescent pregnancy, and violence in the next generation [2]. This has led to global commitments within the Sustainable Development Goals (SDGs) to prevent and respond to violence against children and to the development led by the World Health Organization (WHO) in 2016 of a set of seven strategies for achieving this—INSPIRE.

INSPIRE was launched alongside the Global Partnership to End Violence Against Children by WHO, United States Centers for Disease Control and Prevention (CDC), the Pan American Health Organization (PAHO), President's Emergency Plan for AIDS Relief (PEPFAR), Together for Girls, United Nations International Children's Fund (UNICEF), UN Office on Drugs and Crime (UNODC), US Agency for International Development (USAID), and the

World Bank. It includes seven strategies for ending violence against children: **I**mplementation and enforcement of laws, **N**orms and values, **S**afe Environments, **P**arent and caregiver support, **I**ncome and economic strengthening, **R**esponse and support services, and **E**ducation and life skills (Fig 1) [3]. These strategies are based on reviews of all effective interventions in the growing evidence base for prevention and response to violence against children [4], including youth violence and sexual abuse [5]. INSPIRE has gained widespread uptake globally, with commitments from 27 'pathfinding' countries.

However, implementing the full range of INSPIRE strategies and approaches is a challenge in contexts of limited financial and human resources. This may become even more

| Partners | Centers for Disease Control and Prevention (CDC), the Pan American Health Organization (PAHO), the President's Emergency Program for AIDS Relief (PEPFAR), Together for Girls, the United Nations Children's Fund (UNICEF), United Nations Office on Drugs and Crime (UNODC), United States Agency for International Development (USAID), and the World Bank. |
|---|---|
| **Strategy** | **Approach** |
| 1. Implementation and enforcement of laws | Laws banning violent punishment of children<br>Laws criminalizing sexual abuse and exploitation of children<br>Laws that prevent alcohol misuse<br>Laws limiting youth access to firearms and other weapons |
| 2. Norms and values | Changing adherence to restrictive and harmful gender and social norms<br>Community mobilization programmes<br>Bystander interventions |
| 3. Safe environments | Reducing violence by addressing "hotspots"<br>Interrupting the spread of violence<br>Improving the built environment |
| 4. Parent and caregiver support | Delivered through home visits<br>Delivered in groups in community settings<br>Delivered through comprehensive programmes |
| 5. Income and economic strengthening | Cash transfers<br>Group savings and loans combined with gender equity training<br>Microfinance combined with gender norm training |
| 6. Response and support services | Counselling and therapeutic approaches<br>Screening combined with interventions<br>Treatment programmes for juvenile offenders in the criminal justice system<br>Foster care interventions involving social welfare services |
| 7. Education and life skills | Increase enrolment in pre-school, primary and secondary schools<br>Reduce violence against children by school staff<br>Reduce peer violence and bullying<br>Improve children's knowledge about sexual abuse and how to protect themselves against it<br>Life and social skills training<br>Adolescent intimate partner violence prevention programmes |

**Fig 1. INSPIRE strategies for preventing and responding to violence against children.**

acute with lasting global economic impacts of the COVID-19 epidemic, limiting budgets and services for prevention of violence against children. In parallel, governments, donors, and nongovernmental organisations (NGOs) have committed to a new agenda of integrated and connected SDGs. If we are to meet these commitments, we need to address multiple forms of violence—across several SDG targets—simultaneously. Consequently, governments are asking for a next step in the evidence base: the identification of solutions that can have impact across not one but multiple childhood violence targets. Such multiple-win actions are described by the UN as 'development accelerators' and have the potential to provide efficiencies for governments [6].

Testing a large number of interventions in isolation and in multiple different combinations in a randomised control trial is neither feasible nor affordable. As a first step, observational data can be used to evaluate associations between community- and family-level protective factors and multiple outcomes. Such evidence could guide future policy or programme selection. In 2019, a pioneering analysis evaluated the association between six protective factors and 11 SDG targets, including emotional and physical violence [7]. However, this study focused on a subpopulation of adolescents living with HIV and lacked power to study rarer forms of violence such as sexual abuse or transactional sexual exploitation. The present study responds to policy requests to identify 'accelerators' for violence prevention and response, by evaluating which INSPIRE-aligned protective factors may act on multiple violence types.

In any analysis of adolescent violence, differences in rates of exposure and ramifications mean that it is important to examine violence prevention and response for girls and boys separately [8]. To this end, we identified and pooled two existing prospective cohorts. Our aim was to guide the choice of 'best buy' INSPIRE provisions for governments. The study objectives were to (1) evaluate the association between INSPIRE-aligned protective factors and multiple violence outcomes and (2) assess whether experiencing multiple INSPIRE-aligned protective factors might be linked to greater reductions in the risk of violence.

## Methods

We use secondary data with self-reported measures to examine how variation in protective factors between individuals is associated with multiple forms of violence. All of the protective factors examined are possible targets of INSPIRE interventions. Identifying which protective factors predict lower rates of violence is intended to inform efforts to identify and prioritise the most efficient subset of interventions. The study setting was South Africa—a country with high levels of government commitment to preventing violence against children but also facing implementation challenges shared by many low- and middle-income contexts: limited service delivery capacity, infrastructure, and financial resources. The study was reported according to the Strengthening the Reporting of Observational Studies in Epidemiology (STROBE) checklist for cohort studies (S1 Checklist) [9].

### Data

Study data were pooled from two large prospective cohorts (Young Carers and Mzantsi Wakho) spread across three South African Provinces: Eastern Cape, Mpumalanga, and Western Cape. The Young Carers and Mzantsi Wakho cohorts were designed to allow data merging, with shared investigators, measures, data collection procedures, and sampling in urban and rural sites. The Young Carers recruitment took place between 1 January 2010 and 7 December 2011, with follow-up between 5 January 2011 and 15 December 2012. Participants included $N$ = 3,515 children and adolescents in Mpumalanga and Western Cape Provinces. In each province, census enumeration areas were randomly selected within one urban and one

rural health district, and all households with a resident 10- to 17-year-old were recruited. The cohort had 97% uptake during recruitment and 97% retention at 12- to 18-month follow-up [10]. The Mzantsi Wakho ('Our South Africa') recruitment took place between 4 March 2014 and 25 September 2015, with follow-up between 10 November 2015 and 5 April 2017. Participants included $N$ = 1,519 children and adolescents in the Eastern Cape Province, living with and without HIV. Across two urban and rural health districts, all children and adolescents (aged 10–19 years) who had ever initiated HIV care in 53 health facilities were recruited, alongside their closest child or adolescent neighbours. The cohort had 90% uptake and 94% retention at 12- to 18-month follow-up (with 2.4% mortality) [11].

For both cohorts, interviews took approximately 45–70 minutes and were conducted in a location of the adolescent's choice. For Young Carers, interviews were administered using self-interviewing on paper forms, and for Mzantsi Wakho, interviews were administered using audio mobile-assisted self-interviewing on electronic tablets. All interviews were supported by trained local interviewers, and the level of assistance was adjusted according to the age and literacy level of participants. Questionnaires were available in the languages of the adolescent's choice, including Xhosa, Swati, Tsonga, Pedi, and English. There was no financial payment for participation, but all participants received certificates and snacks, regardless of whether they completed interviews. All participants were asked whether they would be happy to be contacted or visited again for a follow-up at baseline. For those consenting, telephone numbers, addresses, or GPS location was collected for tracing.

## Ethics

Ethical approvals were gained for Young Carers and Mzantsi Wakho from the University of Oxford (SSD/CUREC2/11-40; SSD/CUREC2/12-21) and University of Cape Town (REC REF: CSSR 389/2009; REC REF CSSR 2013/04) and provincial South African Departments of Health, Basic Education, and Social Development. In both cases, ethical approval included examination of predictors of violence with collected data. Both studies obtained written consent from all adolescents and their primary caregivers at both baseline and follow-up. Owing to varying levels of literacy, informed consent information was read aloud to potential participants. Confidentiality was maintained except when participants disclosed serious risk of harm to themselves or others. In such circumstances, safeguarding processes were followed. For those reporting current abuse, recent rape, or suicidality, immediate responses included support to access postexposure prophylaxis, pregnancy prevention, and child protection measures in conjunction with government services. Findings of the studies are reported back to communities and health facilities in research areas as part of ongoing local knowledge-sharing.

## Adolescent involvement

Children and adolescents were involved in the conceptualisation, design, conduct, and dissemination of these two studies and of this paper. This took place through 11 years of Adolescent Advisory Group weekend workshops in urban and rural areas of South Africa, in which children and adolescents (10–20 years old) engaged in planning research studies, designing content and appearance of questionnaires, planning dissemination, and engaging with the South African National Ministry of Health who attended these workshops. In addition, Young Carers and Mzantsi Wakho questionnaires were piloted with 20 and 25 adolescents, respectively. Adolescents were from local areas involved in the respective studies. All comments were taken into consideration and errors amended.

## Measures

Input to questionnaire design in both studies was given by the South African National Departments of Health, Basic Education, and Social Development; the South African National AIDS Council; UNICEF; PEPFAR; USAID; and local NGOs. All questionnaires are available at www.youngcarers.org.za: Young Carers baseline [12] and follow-up [13] and Mzantsi Wakho baseline [14] and follow-up [15].

**Outcomes.** Six violence outcomes were identified in the data, including five measures of violence victimisation and one of violence perpetration. For the analysis, all six outcomes were coded by collapsing responses to multiple self-reported items into binary indicators of any experience. The six outcomes were (1) sexual abuse, measured as reporting either sexual assault and/or completed rape in the last year using items from the Sexual Victimization module of the Juvenile Victimization Questionnaire [16]; (2) transactional sexual exploitation, measured as reporting any lifetime receipt of money, drinks, clothes, cell phone airtime, a place to stay, lifts in a car/taxi, better marks at school, school fees, food, or anything else for having sex with someone (this question was drawn from the National Survey of Risk Behaviour Amongst Young South Africans [17]); (3) physical abuse, measured as reporting a caregiver or other adult to have either used a stick, belt, or other hard item to hit you or slapped, punched, hit, pinched, or pulled your ear/hair so that it hurt or left marks in the last year, as in UNICEF's Psychosocial Vulnerability and Resilience Measures For National-Level Monitoring of Vulnerable Children [18]; (4) emotional abuse, defined as reporting a caregiver or other adult to have said they would call ghosts or evil spirits or harmful people, said you would be sent away or kicked out of the house, and/or called you dumb, lazy, or other names in the last year, as in the same UNICEF Psychosocial Vulnerability and Resilience Measures [18]; (5) community violence victimisation, defined as reporting having had things stolen in the last year, having ever been hit or attacked outside, or having ever seen someone stabbed or shot, as in the Child Exposure to Community Violence checklist [19] (these three items reflect the most common community traumas in South Africa, as identified by police statistics); (6) youth lawbreaking as reporting hanging around with kids that get in trouble, stealing at home, stealing things from places other than home, fighting a lot, and carrying a gun and/or knife for protection in the past 6 months, as in the Child Behaviour Checklist Delinquency subscale [20]. Items about gang affiliation and carrying of a knife and/or gun were added during questionnaire piloting.

**Hypothesised protective factors for violence.** As data collection was initiated before the publication of the INSPIRE framework in 2016, we retrospectively identified seven protective factors that were aligned with its strategies. (1) We measured positive parenting/caregiving using a continuous sum of items 2, 13, 16, and 27 from the child form of the Alabama Parenting Questionnaire [21], which consider warmth and praise from a primary caregiver (range: 0–16). Given high rates of informal fostering in Southern Africa, we note that 'parents' were defined as any primary caregiver and there was no requirement for a biological relationship. (2) We measured parental/caregiver monitoring and supervision using a continuous sum of items 10, 17, and 19, from the child form of the Alabama Parenting Questionnaire, which include setting rules about coming home in evenings, and knowing who an adolescent is friends with (range: 0–12). (3) We measured food security at home as a binary indicator of 6 or 7 days in the past week with enough food in the home. (4) We measured basic economic security at home as a binary indicator of access to all six of the top socially perceived necessities for youth in the South African Social Attitudes Survey: a doctor when needed, school uniform, basic clothing, toiletries to wash, school equipment, and a pair of shoes [22]. (5) We measured free schooling as a binary indicator of enrolment in a no-fees school, or receipt of a fee exemption. (6) We measured free school meals as a binary indicator of daily free lunches or

breakfasts at school [23]. Exposure to these protective factors was measured as consistent reporting at both baseline and follow-up, based on evidence that such factors need to be predictable and sustained in order to protect children and adolescents in high-risk settings [24, 25]. For this, continuous variables were combined additively. (8) We also describe the prevalence of adolescent engagement with response services for children who are victims of violence, measured as any social or justice service response at follow-up related to emotional, physical, or sexual abuse (Young Carers data) and sexual abuse (Mzantsi Wakho data). Because experiencing study outcomes of abuse was a condition for being asked about access to response services, we were unable to evaluate it as a protective factor.

**Covariates.** Control variables included 10 key sociodemographic variables and violence-associated factors: participant sex, age, HIV status, rural/urban household location, household size, informal/shack housing, maternal orphanhood, paternal orphanhood, and province of residence. Participant HIV status was assessed using self-report (Young Carers data) and clinical records (Mzantsi Wakho data). Baseline measures of self-reported sexual abuse, emotional abuse, physical abuse (all last year), and youth lawbreaking in the last 6 months were also included as covariates in regression models investigating associations between hypothesised protective factors and these forms of violence at follow-up.

## Analysis

Analysis took place in seven steps in Stata 15, all stratified by sex. First, we compared baseline characteristics of participants retained and lost to follow-up. Second, we described baseline and follow-up characteristics of retained participants using count (percent) for binary and categorical variables and mean (standard deviation [SD]) for continuous variables. Third, we evaluated univariable associations between the six hypothesised protective factors and six binary violence outcomes. Fourth, we investigated multivariable associations between hypothesised protective factors and violence outcomes. For this, we used path analysis consisting of six single-outcome multivariable logistic regression models, each regressing one of the six violence outcomes at follow-up on the six hypothesised protective factors, controlling for outcome-specific baseline exposure to violence and for common sociodemographic factors. Missing data were handled using listwise deletion. Fifth, to account for risk of type I error from multiple-hypothesis testing, we adjusted estimated $p$-values using the Benjamini-Hochberg procedure specified with a false discovery rate of 5% [26]. The analysis had six regression models, each containing six predictors of scientific interest. Hence, a family of tests consisted of six values, one for each protective factor, across regression models. Sixth, hypothesised protective factors significantly associated (for either sex or both) with three or more violence outcomes were defined as protective factors that had potential to indicate development 'accelerators'. Seventh, we estimated adjusted probabilities for experiencing each violence outcome under three scenarios: (1) experiencing no 'accelerator' protective factors, (2) experiencing each 'accelerator' protective factor, and (3) experiencing all 'accelerator' protective factors together. Positive parenting and parental monitoring and supervision were the only hypothesised protective factors modelled as continuous scales. For these two variables, adjusted probabilities were estimated at the sample mean value and at their maximum scores (32 for positive parenting and 24 for parental monitoring and supervision). Finally, adjusted risk ratios (ARRs) and adjusted risk differences (ARDs) were used to compare scenarios 2 and 3 relative to scenario 1. All confidence intervals (CIs) are given at 95% confidence level.

Although the specific analysis reported in this study was not prespecified, it followed a prespecified methodological approach that has been developed to investigate factors associated

with multiple SDG outcomes in observational data as part of the UKRI GCRF 'Accelerate Hub' [27]. This approach is laid out on Open Science Framework [28].

### Sensitivity analysis

Multiple-outcome models that include correlation between the error terms of outcomes can be more accurate when modelling highly correlated outcomes with different sets of predictors per outcome [29]. As a sensitivity analysis, we calculated adjusted probabilities of experiencing each violence outcome from a multiple-outcome probit model that correlated the error terms of our six different regression models. This was done using the mvprobit command in Stata 15 set at 100 random draws [30]. As in our main analysis, each regression regressed one of the six violence outcomes at follow-up on the six hypothesised protective factors, controlling for outcome-specific baseline exposure to violence and for common sociodemographic factors.

## Results

### Participants

The pooled data provided a sample of $N = 5,034$ children and adolescents at baseline. Across the two pooled cohorts, median time to follow-up was 353 days, and 4% of participants ($N = 223$) were lost to follow-up. At baseline, participants lost to follow-up were more likely to live in the Eastern Cape Province ($p < 0.001$), be older ($p < 0.001$), be living with HIV ($p < 0.001$), and live in an urban area ($p = 0.002$) (S1 Table). In terms of violence, at baseline, participants lost to follow-up were equally likely to experience sexual abuse, transactional sexual exploitation, physical abuse, and youth lawbreaking but more likely to experience emotional abuse ($p = 0.022$) and community violence victimisation ($p = 0.012$) (S1 Table). The final analysis included 4,811 participants. Missing values for all variables were <1%.

### Descriptive data

Baseline and follow-up characteristics of study participants are summarised in Table 1. Fifty-six percent of participants were female. At baseline, the average age was 13.54 years (SD: 2.41 years), 19% of participants were maternally orphaned, 23% paternally orphaned, 7% doubly orphaned, and 22% living with HIV. Mean household size including the participant was 5.72 (SD: 2.56). Forty-three percent lived in rural areas and 28% in informal shacks. Girls were slightly older than boys (13.68 years compared to 13.36 years, $p < 0.001$) and more likely to live in informal housing ($p = 0.03$) and households with a larger number of persons ($p < 0.001$). Baseline comparison of study participants by province is reported in S3 Table.

Consistent exposure to hypothesised protective factors at both baseline and follow-up was most common for free school meals (80%), food security at home (64%), free schooling (39%), and basic economic security at home (23%). The mean score of parental monitoring and supervision across baseline and follow-up was 19.53 (SD: 3.89), and 22% reported the maximum score of 24.00. The mean score of positive parenting was 24.51 (SD: 5.89), and 16% reported the maximum score of 32.00. Girls were more likely to experience better parental monitoring and supervision ($p < 0.001$) but had less food security at home ($p < 0.001$) or free school meals ($p = 0.019$). Access to any kind of support service in response to abuse was very rare (3% of those reporting abuse).

At follow-up, the most common form of reported violence was community violence victimisation (37%), followed by physical abuse (33%), youth lawbreaking (23%), emotional abuse (21%), transactional sexual exploitation (8%), and sexual abuse (3%). Girls were more likely to experience sexual abuse, transactional sexual exploitation, and emotional abuse ($p < 0.001$),

**Table 1. Characteristics of study participants.**

| | Baseline | | | | Follow-up | | | |
|---|---|---|---|---|---|---|---|---|
| | Total *n* = 4,811 | Boys *n* = 2,087 | Girls *n* = 2,724 | Sex difference *p*-value | Total *n* = 4,811 | Boys *n* = 2,087 | Girls *n* = 2,724 | Sex difference *p*-value |
| **Sociodemographic characteristics** | | | | | | | | |
| Province | | | | 0.096 | | | | 0.096 |
| Eastern Cape | 1,410 (29) | 612 (29) | 798 (29) | | 1,410 (29) | 612 (29) | 798 (29) | |
| Western Cape | 1,753 (36) | 729 (34) | 1,024 (37) | | 1,753 (36) | 729 (35) | 1,024 (38) | |
| Mpumalanga | 1,648 (34) | 746 (35) | 902 (33) | | 1,648 (34) | 746 (36) | 902 (33) | |
| Age | | | | <0.001 | | | | <0.001 |
| Mean (SD) | 13.54 (2.41) | 13.36 (2.27) | 13.68 (2.51) | | 14.85 (2.47) | 14.68 (2.34) | 14.99 (2.57) | |
| Maternal orphan | | | | 0.064 | | | | 0.09 |
| Yes | 913 (19) | 421 (20) | 492 (18) | | 1,060 (22) | 484 (23) | 576 (21) | |
| Paternal orphan | | | | 0.168 | | | | 0.40 |
| Yes | 1,090 (23) | 453 (22) | 637 (23) | | 1,321 (27) | 560 (27) | 761 (28) | |
| Living with HIV | | | | 0.388 | | | | 0.47 |
| Yes | 1,042 (22) | 464 (22) | 578 (21) | | 1,058 (22) | 469 (22) | 589 (22) | |
| Rural location | | | | 0.086 | | | | 0.083 |
| Yes | 2,067 (43) | 867 (42) | 1,200 (44) | | 2,048 (43) | 859 (41) | 1,189 (44) | |
| Informal housing | | | | 0.03 | | | | 0.058 |
| Yes | 1,322 (28) | 540 (26) | 782 (29) | | 1,257 (26) | 518 (25) | 739 (27) | |
| Household size | | | | <0.001 | | | | 0.005 |
| Mean (SD) | 5.72 (2.56) | 5.58 (2.44) | 5.83 (2.64) | | 5.44 (2.74) | 5.32 (2.53) | 5.54 (2.88) | |
| **Hypothesised protective factors for violence**[*] | | | | | | | | |
| Positive parenting | | | | 0.784 | | | | 0.664 |
| Mean (SD) | 12.27 (3.68) | 12.29 (3.61) | 12.26 (3.74) | | 24.51 (5.89) | 24.55 (5.71) | 24.48 (6.02) | |
| Parental monitoring and supervision | | | | 0.001 | | | | <0.001 |
| Mean (SD) | 9.79 (2.62) | 9.65 (2.66) | 9.90 (2.59) | | 19.53 (3.89) | 19.19 (3.98) | 19.79 (3.80) | |
| Food security at home | | | | 0.002 | | | | <0.001 |
| Yes | 3,782 (79) | 1,684 (81) | 2,098 (77) | | 3,101 (64) | 1,421 (68) | 1,680 (62) | |
| Basic economic security at home | | | | 0.416 | | | | 0.177 |
| Yes | 1,960 (41) | 865 (41) | 1,095 (40) | | 1,112 (23) | 503 (24) | 609 (22) | |
| Free schooling | | | | 0.318 | | | | 0.533 |
| Yes | 2,338 (49) | 997 (48) | 1,341 (49) | | 1,892 (39) | 810 (39) | 1,082 (40) | |
| Free school meals | | | | 0.566 | | | | 0.019 |
| Yes | 4,149 (86) | 1,807 (87) | 2,342 (86) | | 3,847 (80) | 1,703 (82) | 2,144 (79) | |
| Abuse response services[†] | | | | - | | | | <0.001 |
| Yes | - | - | - | | 133 (3) | 31 (1) | 102 (4) | |
| **Violence outcomes** | | | | | | | | |
| Sexual abuse | | | | <0.001 | | | | <0.001 |
| Yes | 180 (4) | 45 (2) | 135 (5) | | 163 (3) | 49 (2) | 114 (4) | |
| Transactional sexual exploitation | | | | <0.001 | | | | <0.001 |
| Yes | 190 (4) | 44 (2) | 146 (5) | | 381 (8) | 133 (6) | 248 (9) | |
| Physical abuse | | | | 0.849 | | | | 0.365 |
| Yes | 1,648 (34) | 718 (34) | 930 (34) | | 1,595 (33) | 707 (34) | 888 (33) | |
| Emotional abuse | | | | 0.002 | | | | <0.001 |
| Yes | 1,343 (28) | 536 (26) | 807 (30) | | 1,023 (21) | 394 (19) | 629 (23) | |

(*Continued*)

**Table 1.** (Continued)

| | Baseline | | | | Follow-up | | | |
|---|---|---|---|---|---|---|---|---|
| | Total<br>$n = 4,811$ | Boys<br>$n = 2,087$ | Girls<br>$n = 2,724$ | Sex difference $p$-value | Total<br>$n = 4,811$ | Boys<br>$n = 2,087$ | Girls<br>$n = 2,724$ | Sex difference $p$-value |
| Community violence victimisation | | | | <0.001 | | | | <0.001 |
| Yes | 1,965 (41) | 911 (44) | 1,054 (39) | | 1,757 (37) | 826 (40) | 931 (34) | |
| Youth lawbreaking | | | | <0.001 | | | | <0.001 |
| Yes | 1,297 (27) | 648 (31) | 649 (24) | | 1,089 (23) | 565 (27) | 524 (19) | |

Data are mean (SD) for continuous variables and $n$ (%) for categorical variables.

*Hypothesised protective factors at follow-up were measured as consistent access at both baseline and follow-up. For example, 1,684 boys experienced food security at T1, of whom 1,421 also experienced it at T2.

†Measured in the Young Carers data as any social or justice service response at follow-up related to emotional, physical, or sexual abuse and in Mzantsi Wakho as accessing any social or justice services at follow-up related to sexual abuse.

Abbreviation: SD, standard deviation

whereas boys were more likely to experience community violence victimisation, and youth lawbreaking ($p < 0.001$).

## Multivariable associations between hypothesised protective factors and violence outcomes

Of the six hypothesised protective factors from the INSPIRE framework, three factors—positive parenting, parental monitoring and supervision, and food security—were significantly associated with lower likelihood of three or more violence outcomes at follow-up for either girls or boys (Table 2). For girls, each unit increase in positive parenting was associated with lower odds of sexual abuse (AOR: 0.95, 95% CI 0.92–0.98, $p < 0.001$), physical abuse (AOR: 0.98, 95% CI 0.97–0.99, $p = 0.009$), emotional abuse (AOR: 0.96, 95% CI 0.95–0.98, $p < 0.001$), and youth lawbreaking (AOR 0.96, 95% CI 0.95–0.98, $p < 0.001$); each unit increase in parental monitoring and supervision was associated with lower odds of sexual abuse (AOR: 0.94, 95% CI 0.90–0.99, $p = 0.02$), transactional sexual exploitation (AOR: 0.94, 95% CI 0.91–0.97, $p < 0.001$), physical abuse (AOR: 0.95, 95% CI 0.93–0.97, $p < 0.001$), emotional abuse (AOR: 0.94, 95% CI 0.92–0.97, $p < 0.001$), community violence victimisation (AOR: 0.97, 95% CI 0.94–0.99, $p = 0.012$), and youth lawbreaking (AOR: 0.94, 95% CI 0.92–0.97, $p < 0.001$); food security at home was associated with lower odds of sexual abuse (AOR: 0.54, 95% CI 0.35–0.81, $p = 0.003$), transactional sexual exploitation (AOR: 0.55, 95% CI 0.40–0.76, $p < 0.001$), physical abuse (AOR: 0.70, 95% CI 0.58–0.84, $p < 0.001$), emotional abuse (AOR: 0.74, 95% CI 0.61–0.91, $p = 0.004$), and community violence victimisation (AOR: 0.73, 95% CI 0.60–0.89, $p = 0.002$). For boys, each unit increase in positive parenting was associated with lower odds of emotional abuse (AOR: 0.98, 95% CI 0.96–1.00, $p = 0.028$) and youth lawbreaking (AOR: 0.97, 95% CI 0.95–0.99, $p = 0.001$); each unit increase in parental monitoring and supervision was associated with lower odds of transactional sexual exploitation (AOR: 0.93, 95% CI 0.89–0.97, $p = 0.001$), physical abuse (AOR: 0.96, 95% CI 0.94–0.99, $p = 0.005$), emotional abuse (AOR: 0.94, 95% CI 0.91–0.97, $p < 0.001$), community violence victimisation (AOR: 0.94, 95% CI 0.92–0.97, $p < 0.001$), and youth lawbreaking (AOR: 0.91, 95% CI 0.88–0.93, $p < 0.001$); food security at home was associated with lower odds of physical abuse (AOR: 0.72, 95% CI 0.58–0.90, $p = 0.003$) and emotional abuse (AOR: 0.59, 95% CI 0.46–0.76, $p < 0.001$). Univariable associations between hypothesised protective factors and violence outcomes are reported in S3 Table.

**Table 2. Summary of multivariable associations between hypothesised protective factors and violence outcomes.**

| | Boys | | Girls | | Sex interaction |
|---|---|---|---|---|---|
| | AOR (95% CI) | *p*-value | AOR (95% CI) | *p*-value | *p*-value |
| **Sexual abuse** | | | | | |
| Positive parenting | 0.97 (0.92–1.02) | 0.176 | 0.95 (0.92–0.98) | <0.001* | 0.61 |
| Parental monitoring and supervision | 0.99 (0.92–1.07) | 0.83 | 0.94 (0.90–0.99) | 0.02 | 0.27 |
| Food security at home | 1.00 (0.50–2.01) | 0.999 | 0.54 (0.35–0.81) | 0.003* | 0.13 |
| Basic economic security at home | 1.13 (0.56–2.27) | 0.74 | 0.84 (0.48–1.46) | 0.538 | 0.52 |
| Free schooling | 1.99 (1.00–3.97) | 0.051 | 1.27 (0.79–2.05) | 0.327 | 0.27 |
| Free school meals | 0.51 (0.24–1.09) | 0.084 | 0.96 (0.58–1.60) | 0.872 | 0.16 |
| **Transactional sexual exploitation** | | | | | |
| Positive parenting | 0.98 (0.95–1.02) | 0.359 | 0.99 (0.97–1.02) | 0.56 | 0.69 |
| Parental monitoring and supervision | 0.93 (0.89–0.97) | 0.001* | 0.94 (0.91–0.97) | <0.001* | 0.63 |
| Food security at home | 0.94 (0.62–1.45) | 0.792 | 0.55 (0.40–0.76) | <0.001* | 0.05 |
| Basic economic security at home | 0.71 (0.42–1.21) | 0.205 | 0.88 (0.56–1.36) | 0.561 | 0.54 |
| Free schooling | 1.00 (0.65–1.56) | 0.985 | 0.54 (0.37–0.81) | 0.002* | 0.04 |
| Free school meals | 0.68 (0.42–1.11) | 0.12 | 0.84 (0.58–1.21) | 0.343 | 0.47 |
| **Physical abuse** | | | | | |
| Positive parenting | 0.99 (0.97–1.00) | 0.135 | 0.98 (0.97–0.99) | 0.009* | 0.59 |
| Parental monitoring and supervision | 0.96 (0.94–0.99) | 0.005* | 0.95 (0.93–0.97) | <0.001* | 0.41 |
| Food security at home | 0.72 (0.58–0.90) | 0.003* | 0.70 (0.58–0.84) | <0.001* | 0.78 |
| Basic economic security at home | 1.15 (0.90–1.46) | 0.274 | 1.00 (0.80–1.25) | 0.98 | 0.42 |
| Free schooling | 1.25 (1.01–1.54) | 0.042 | 1.06 (0.87–1.28) | 0.574 | 0.24 |
| Free school meals | 1.32 (1.00–1.74) | 0.053 | 1.29 (1.02–1.64) | 0.032 | 0.92 |
| **Emotional abuse** | | | | | |
| Positive parenting | 0.98 (0.96–1.00) | 0.028* | 0.96 (0.95–0.98) | <0.001* | 0.26 |
| Parental monitoring and supervision | 0.94 (0.91–0.97) | <0.001* | 0.94 (0.92–0.97) | <0.001* | 0.90 |
| Food security at home | 0.59 (0.46–0.76) | <0.001* | 0.74 (0.61–0.91) | 0.004* | 0.16 |
| Basic economic security at home | 1.20 (0.88–1.62) | 0.247 | 0.84 (0.64–1.10) | 0.199 | 0.08 |
| Free schooling | 1.34 (1.04–1.73) | 0.022 | 0.94 (0.76–1.16) | 0.555 | 0.03 |
| Free school meals | 0.88 (0.64–1.22) | 0.444 | 1.02 (0.79–1.32) | 0.854 | 0.46 |
| **Community violence victimisation** | | | | | |
| Positive parenting | 1.00 (0.98–1.02) | 0.861 | 1.00 (0.98–1.01) | 0.707 | 0.70 |
| Parental monitoring and supervision | 0.94 (0.92–0.97) | <0.001* | 0.97 (0.94–0.99) | 0.012* | 0.18 |
| Food security at home | 0.91 (0.72–1.16) | 0.458 | 0.73 (0.60–0.89) | 0.002* | 0.15 |
| Basic economic security at home | 0.76 (0.57–1.01) | 0.058 | 0.85 (0.64–1.14) | 0.276 | 0.56 |
| Free schooling | 0.86 (0.68–1.09) | 0.215 | 1.16 (0.94–1.44) | 0.175 | 0.06 |
| Free school meals | 0.76 (0.56–1.03) | 0.076 | 0.83 (0.63–1.08) | 0.167 | 0.65 |
| **Youth lawbreaking** | | | | | |
| Positive parenting | 0.97 (0.95–0.99) | 0.001* | 0.96 (0.95–0.98) | <0.001* | 0.73 |
| Parental monitoring and supervision | 0.91 (0.88–0.93) | <0.001* | 0.94 (0.92–0.97) | <0.001* | 0.04 |
| Food security at home | 1.19 (0.94–1.51) | 0.141 | 0.96 (0.77–1.19) | 0.71 | 0.18 |
| Basic economic security at home | 0.92 (0.70–1.20) | 0.526 | 0.80 (0.60–1.07) | 0.132 | 0.50 |
| Free schooling | 0.64 (0.50–0.80) | <0.001* | 0.64 (0.51–0.81) | <0.001* | 0.98 |
| Free school meals | 1.05 (0.79–1.40) | 0.730 | 1.08 (0.83–1.41) | 0.583 | 0.90 |

With listwise deletion $N$ = 4,641. AORs and 95% CIs were estimated jointly in a single multivariate multivariable path model. Associations between hypothesised protective factors and the six outcomes are mutually adjusted for all other hypothesised protective factors and for participant sociodemographic characteristics: age, maternal orphanhood, paternal orphanhood, HIV status, number of people living in household, rural/urban household location, informal housing type (shack), and province of residence. Associations between hypothesised protective factors and sexual abuse, emotional abuse, physical abuse, and youth lawbreaking are also adjusted for self-reported exposure to these forms of violence at baseline.

*p-Values are significant after applying the Benjamini-Hochberg procedure specified with a false discovery rate of 5%.

Abbreviations: AOR, adjusted odds ratio; CI, confidence interval

## Adjusted probability of violence outcomes comparing different combinations of 'accelerator' protective factors

For girls, the adjusted probability of all six types of violence was estimated to be significantly lower for a scenario in which high positive parenting, high parental monitoring and supervision, and food security at home were experienced together as compared to a scenario of average positive parenting, average parental monitoring, and no food security at home (Table 3 and Fig 2A). With no protective factors, the adjusted probability of sexual abuse was 5.38%; with all three, it was 1.64% (ARD: −3.74% points, 95% CI −5.31 to −2.16, $p < 0.001$). With no protective factors, the adjusted probability of transactional sexual exploitation was 10.07%; with all three, it was 4.84% (ARD: −5.23% points, 95% CI −7.26 to −3.20, $p < 0.001$). With no protective factors, the adjusted probability of physical abuse was 38.58; with all three, it was 23.85% (ARD: −14.72% points, 95% CI −19.11 to −10.53, $p < 0.001$). With no protective factors, the adjusted probability of experiencing emotional abuse was 25.39%; with all three, it was 12.98% (ARD: −12.41% points, 95% CI −16.00 to −8.83, $p < 0.001$). With no protective factors, the adjusted probability of community violence victimisation was 36.25%; with all three, it was 28.37% (ARD: −7.87% points, 95% CI −11.98 to −3.76, $p < 0.001$). With no protective factors, the adjusted probability of youth lawbreaking was 18.90%; with all three, it was 11.61% (ARD: −7.30% points, 95% CI −10.50 to −4.09, $p < 0.001$).

For boys, the adjusted probability of transactional sexual exploitation, physical abuse, emotional abuse, community violence victimisation, and youth lawbreaking was estimated to be significantly lower for a scenario in which high positive parenting, high parental monitoring and supervision, and food security at home were experienced together as compared to a scenario of average positive parenting, average parental monitoring, and no food security at home (Table 3 and Fig 2B). With no protective factors, the adjusted probability of engaging in transactional sexual exploitation was 6.97%; with all three, it was 4.55% (ARD: −2.42% points, 95% CI −4.77 to −0.08, $p = 0.043$). With no protective factors, the adjusted probability of experiencing physical abuse was 37.19%; with all three, it was 25.44% (ARD: −11.74% points, 95% CI −16.91 to −6.58, $p < 0.001$). With no protective factors, the adjusted probability of experiencing emotional abuse was 23.72%; with all three, it was 10.72% (ARD: −13.00% points, 95% CI −17.04 to −8.95, $p < 0.001$). With no protective factors, the adjusted probability of experiencing community violence victimisation was 41.28%; with all three, it was 35.41% (ARD: −5.87% points, 95% CI −10.98 to −0.75, $p = 0.025$). With no protective factors, the adjusted probability of youth lawbreaking was 22.44%; with all three, it was 14.98% (ARD: −7.46% points, 95% CI −11.57 to −3.35, $p < 0.001$).

## Sensitivity analysis

Accounting for correlation between our outcomes using a multiple-outcome model showed equivalent results, with no differences in significant findings between the two methods. However, the combination of positive parenting, parental monitoring and supervision, and food security was associated with even lower adjusted probability of community violence victimisation in both sexes (ARD: −8.78% points in boys and −8.22% points in girls) (S4 Table).

## Discussion

Findings from our analysis provide important new evidence on the association between protective factors and multiple violence outcomes. They suggest that food security, positive parenting, and parental monitoring and supervision may be associated with reduced risk of three or more violence outcomes related to SDGs 5 'Gender equality' and 16 'Peace, justice, and

**Table 3. Summary of adjusted probabilities, risk ratios, and risk differences for violence outcomes.**

| | Boys | | | Girls | | |
|---|---|---|---|---|---|---|
| | Adjusted probability, %[*] | Exposure versus no exposure | | Adjusted probability, %[*] | Exposure versus no exposure | |
| | | ARR (95% CI) | ARD (95% CI) | | ARR (95% CI) | ARD (95% CI) |
| **Sexual abuse** | | | | | | |
| No protective factors | 2.39 | - | - | 5.38 | - | - |
| Positive parenting | 1.86 | 0.78 (0.50–1.06) | −0.53 (−1.24 to 0.19) | 3.81 | 0.71 (0.56–0.86) | −1.57 (−2.43 to −0.71) |
| Parental monitoring and supervision | 2.31 | 0.97 (0.66–1.27) | −0.08 (−0.81 to 0.64) | 4.24 | 0.79 (0.63–0.95) | −1.14 (−2.01 to −0.26) |
| Food security at home | 2.39 | 1.00 (0.33–1.67) | 0.00 (−1.59 to 1.59) | 3.00 | 0.56 (0.34–0.78) | −2.38 (−4.09 to −0.67) |
| All three protective factors | 1.80 | 0.75 (0.16–1.34) | −0.59 (−2.24 to 1.05) | 1.64 | 0.30 (0.16–0.45) | −3.74 (−5.31 to −2.16) |
| **Transactional sexual exploitation** | | | | | | |
| No protective factors | 6.97 | - | - | 10.07 | - | - |
| Positive parenting | 6.35 | 0.91 (0.73–1.09) | −0.63 (−1.91 to 0.66) | 9.67 | 0.96 (0.83–1.09) | −0.40 (−1.73 to 0.93) |
| Parental monitoring and supervision | 5.26 | 0.76 (0.63–0.88) | −1.71 (−2.55 to −0.86) | 8.15 | 0.81 (0.71–0.91) | −1.92 (−2.94 to −0.91) |
| Food security at home | 6.66 | 0.95 (0.63–1.28) | −0.31 (−2.67 to 2.04) | 6.38 | 0.63 (0.48–0.79) | −3.70 (−5.73 to −1.66) |
| All three protective factors | 4.55 | 0.65 (0.38–0.93) | −2.42 (−4.77 to −0.08) | 4.84 | 0.48 (0.33–0.63) | −5.23 (−7.26 to −3.20) |
| **Physical abuse** | | | | | | |
| No protective factors | 37.19 | - | - | 38.58 | - | - |
| Positive parenting | 35.00 | 0.94 (0.86–1.02) | −2.19 (−5.01 to 0.64) | 35.33 | 0.92 (0.85–0.98) | −3.25 (−5.63 to −0.87) |
| Parental monitoring and supervision | 33.71 | 0.91 (0.84–0.97) | −3.48 (−5.80 to −1.15) | 33.70 | 0.87 (0.82–0.93) | −4.88 (−7.04 to −2.71) |
| Food security at home | 30.44 | 0.82 (0.71–0.93) | −6.75 (−11.31 to −2.19) | 30.85 | 0.80 (0.71–0.89) | −7.73 (−11.63 to −3.82) |
| All three protective factors | 25.44 | 0.68 (0.57–0.80) | −11.74 (−16.91 to −6.58) | 23.85 | 0.62 (0.52–0.71) | −14.72 (−19.11 to −10.33) |
| **Emotional abuse** | | | | | | |
| No protective factors | 23.72 | - | - | 25.39 | - | - |
| Positive parenting | 20.78 | 0.88 (0.77–0.98) | −2.93 (−5.43 to −0.44) | 20.51 | 0.81 (0.73–0.88) | −4.88 (−6.83 to −2.93) |
| Parental monitoring and supervision | 19.25 | 0.81 (0.73–0.90) | −4.47 (−6.42 to −2.51) | 20.86 | 0.82 (0.75–0.89) | −4.53 (−6.36 to −2.70) |
| Food security at home | 15.68 | 0.66 (0.53–0.79) | −8.04 (−12.07 to −4.01) | 20.27 | 0.80 (0.68–0.92) | −5.12 (−8.62 to −1.61) |
| All three protective factors | 10.72 | 0.45 (0.33–0.57) | −13.00 (−17.04 to −8.95) | 12.98 | 0.51 (0.40–0.62) | −12.41 (−16.00 to −8.83) |
| **Community violence victimisation** | | | | | | |
| No protective factors | 41.28 | - | - | 36.25 | - | - |
| Positive parenting | 41.51 | 1.01 (0.94–1.07) | 0.23 (−2.36 to 2.82) | 35.84 | 0.99 (0.93–1.05) | −0.40 (−2.51 to 1.70) |
| Parental monitoring and supervision | 36.72 | 0.89 (0.84–0.94) | −4.56 (−6.66 to −2.45) | 33.79 | 0.93 (0.88–0.99) | −2.46 (−4.36 to −0.55) |
| Food security at home | 39.69 | 0.96 (0.86–1.06) | −1.59 (−5.81 to 2.63) | 31.05 | 0.86 (0.77–0.94) | −5.20 (−8.54 to −1.86) |
| All three protective factors | 35.41 | 0.86 (0.74–0.97) | −5.87 (−10.98 to −0.75) | 28.37 | 0.78 (0.68–0.89) | −7.87 (−11.98 to −3.76) |

(*Continued*)

**Table 3.** (Continued)

| | Boys | | | Girls | | |
|---|---|---|---|---|---|---|
| | Adjusted probability, %* | Exposure versus no exposure | | Adjusted probability, %* | Exposure versus no exposure | |
| | | ARR (95% CI) | ARD (95% CI) | | ARR (95% CI) | ARD (95% CI) |
| **Youth lawbreaking** | | | | | | |
| No protective factors | 22.44 | - | - | 18.90 | - | - |
| Positive parenting | 18.62 | 0.83 (0.74–0.92) | −3.82 (−5.90 to −1.74) | 15.14 | 0.80 (0.72–0.89) | −3.76 (−5.36 to −2.16) |
| Parental monitoring and supervision | 15.84 | 0.71 (0.63–0.78) | −6.61 (−8.21 to −5.00) | 15.17 | 0.80 (0.72–0.88) | −3.74 (−5.26 to −2.21) |
| Food security at home | 25.52 | 1.14 (0.94–1.33) | 3.08 (−0.96 to 7.12) | 18.30 | 0.97 (0.80–1.13) | −0.60 (−3.78 to 2.58) |
| All three protective factors | 14.98 | 0.67 (0.51–0.82) | −7.46 (−11.57 to −3.35) | 11.61 | 0.61 (0.48–0.75) | −7.30 (−10.50 to −4.09) |

*Adjusted probabilities were estimated with basic economic security at home, free schooling, free school meals, and all other covariates at observed values. Not experiencing positive parenting or parental monitoring and supervision was considered to be equivalent to the sample mean score of these factors (24.51 for positive parenting and 19.53 for parental monitoring and supervision). Experiencing these protective factors was considered to be equivalent to experiencing the maximum score of these factors (32.00 for positive parenting and 24.00 for parental monitoring and supervision).

Abbreviations: ARD, adjusted risk difference; ARR, adjusted risk ratio; CI, confidence interval

strong institutions'. Results also indicate that experiencing all three of these factors may be associated with lower risk of violence as compared to experiencing any one of them alone. The large sample used in this analysis enabled disaggregation by sex, and investigation of underreported forms of violence including sexual abuse and transactional sexual exploitation. Although some differences by sex reflect distinct experiences and prevalence of violence victimisation, there was strong similarity in protective factors across sexes. Food security and positive and supervisory caregiving were shared 'accelerators' across girls and boys—suggesting that the same protective factors can be beneficial for both sexes. This is especially important when considering implications of these findings for future interventions. Provisions to support food security and parenting primarily act at a household level, where targeting of one sex can be problematic. Evidence of protective associations for both boys and girls can increase both the value and feasibility of delivering INSPIRE-associated provisions. It is notable that in girls, free schooling was also significantly associated with lower odds of transactional sexual exploitation and youth lawbreaking.

To our knowledge, this study provides the first evidence on which protective factors may be targeted to achieve improvements across diverse violence types simultaneously. Although desirable, it would have been too costly to test all protective factors, and their various combinations, using experimental methods. Instead, we took advantage of observed variation between individuals to provide critical guidance on which protective factors could be considered as effective targets for governments seeking efficiencies in the delivery of INSPIRE. A drawback of this observational approach is that it is unable to say whether identified relationships between self-reported protective factors and violence outcomes are causal. In relation to existing literature, study findings are supported by evidence that positive and supervisory caregiving are important mediators of the effects of parenting support interventions on physical abuse, emotional abuse, and behaviour problems [31]. They are also consistent with growing observational evidence linking parenting to adolescent violence victimisation and perpetration across settings [32]. With regards to food security, a recent systematic review found that cash transfer interventions, which are known to increase food security, are linked to reduced risk of

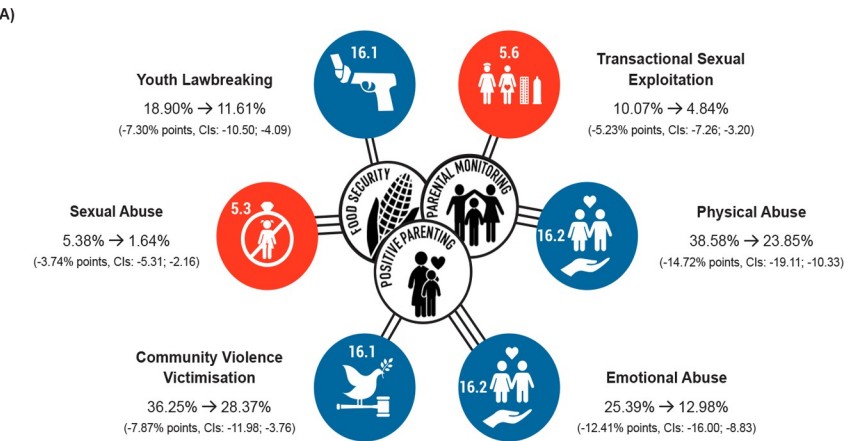

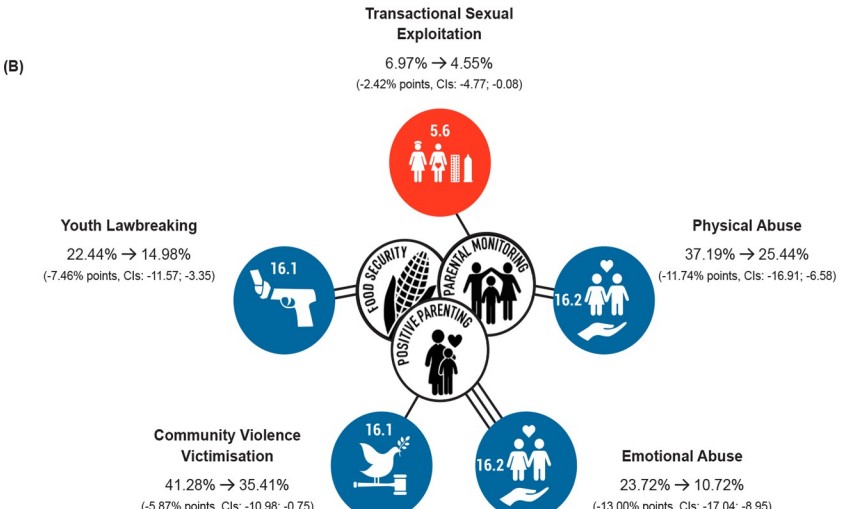

**Fig 2. Adjusted probabilities and risk differences for experiencing violence outcomes in (A) girls and (B) boys.**
Percentages joined by an arrow are adjusted probabilities for the scenarios (1) experiencing no 'accelerator' protective factors and (2) experiencing all three 'accelerator' protective factors. Data in the brackets are adjusted risk differences, and 95% CIs. Double lines between central and outer circles indicate an additive effect of two accelerators; triple lines indicate an additive effect of three accelerators. Blue circles correspond to outcomes related to SDG 16: Peace, Justice, and Strong Institutions. Red circles correspond to outcomes related to SDG 5: Gender Equality. Adjusted probabilities were estimated with basic economic security at home, free schooling, free school meals, and all other covariates at observed values. CI, confidence interval; SDG, Sustainable Development Goal.

sexual abuse amongst female adolescents [33, 34]. Following up on this review, a more recent study in Zimbabwe identified food security as a potential mediator of the effect of economic strengthening on physical violence among adolescents [35]. Food security may reduce pressure on adolescent girls to work outside the home or empower them to leave exploitative relationships [36].

Although our study is unable to establish whether estimated associations are causal, rigorous research methods ensure the highest possible internal validity. The research was conducted using strong longitudinal data, linking repeatedly measured protective factors to prospective violence outcomes. Both cohorts had high rates of uptake (>90%) and retention (>94%) and

minimal missing data. Risk of reverse causality is further reduced by evaluation of protective factors reflecting circumstances external to adolescent participants. The study also adjusts for the common confounding variables including participant sex, age, caregiver status, and socio-economic status. A threat to internal validity is risk of residual confounding from unmeasured variables such as caregiver attitudes and knowledge, and future research should consider this further. Generalisability of study findings is supported by both strong internal validity as well as evidence of the role of parent-child relationships and poverty in determining violence against adolescents across diverse settings and groups in Africa [37, 38]. Known contextual differences in the prevalence of protective factors and violence, as well as the interplay between them, mean that further work is needed to confirm this in other regions [39].

The study has a number of limitations. Because the data used in this analysis were collected prior to the publication of the INSPIRE framework, it did not include potential protective factors within the 'Norms and Values' or 'Safe Environments' strategies. Our measure of access to abuse response services was also not perfectly matched across our two cohorts. Despite this, the study highlights that the proportion of victims reaching any services may be extremely small (<3%). Indeed, such negligible access rates reflect an important need for additional focus on these services. As with all violence research in low-resource settings, this study used self-reported violence outcomes and therefore risks underreporting, particularly of outcomes such as sexual abuse. Evaluating use of justice or social service records is difficult in contexts of very limited service availability. We attempted to reduce bias by using questions previously validated in this population and setting, and by building trust with respondents. For transactional sexual exploitation, participants were asked about ever experiencing this outcome, and it is possible that household food security and economic security were influenced by exposure to transactional sexual exploitation prior to baseline. Any measurement error from this is expected to be minimal, as qualitative data from the region stress the financial and emotional dependence of adolescents on transactional sexual relationships once initiated [40]. In the Young Carers data, HIV status was only measured at follow-up, which may have introduced some measurement error. Considering the expected rate of new infections over the study period, this will have been minimal.

This study provides some of the first evidence, to our knowledge, on which protective factors could be effective targets for interventions aiming to address multiple violence outcomes simultaneously. Each of the self-reported protective factors found to be associated with multiple violence outcomes have corresponding interventions with randomised evidence of effectiveness in low- and middle-income countries. Parenting support programmes improve both positive and supervisory caregiving [41], and household food security may be increased by either cash transfers or other nutrition-sensitive programmes [33, 42]. Combining parenting support with economic strengthening is also an effective way to address these household-level factors in an integrated way [43–46]. Before strong policy recommendations can be made, a number of further research steps are necessary. Additional analysis is needed to disaggregate within- and between-person effects of protective factors and thus strengthen causal claim. Such research should aim to explore hypothesised protective factors across the full spectrum of recommended INSPIRE strategies and focus on a range of settings where the interplay between family, community, or cultural factors and violence may differ from South Africa. Randomised control trials are considered the 'gold-standard' approach for investigating causal effects but, as discussed, can be prohibitively complex and expensive when considering multiple combinations of distinct treatment components. As evidence linking intermediary protective factors to multiple violence outcomes grows, and a reduced set of intervention targets emerges, experimental evaluation of interventions designed to impact on these factors will be

necessary. It would also be valuable to consider if and how services known to modify these protective factors could be implemented at scale effectively and affordably.

The INSPIRE framework has been important in vision and impact: identifying the best and most recent evidence, intensifying the focus on evidence-based programmes, and increasing country-level commitments to prevent violence against children. But for many governments, delivering the full package is challenged by limited resources. These findings suggest a valuable additional step in the process from evidence review to implementation. By identifying INSPIRE-aligned protective factors associated with multiple violence outcomes, it may be possible to pinpoint interventions that could bring efficiency gains and contribute to our growing understanding of 'development accelerators'. Within the broader development agenda of interconnected Global Goals, we are moving towards a new generation of violence prevention programming that is not only effective but also scalable.

## Supporting information

**S1 Checklist. Study STROBE Statement.** STROBE, Strengthening the Reporting of Observational Studies in Epidemiology.
(DOCX)

**S1 Table. Baseline characteristics of study participants by loss to follow-up. Data are mean (SD) for continuous variables and *n* (%) for categorical variables. SD, standard deviation.**
(DOCX)

**S2 Table. Baseline characteristics of study participants by province of residence.** Data are mean (SD) for continuous variables and *n* (%) for categorical variables. SD, standard deviation.
(DOCX)

**S3 Table. Summary of univariable associations between hypothesised protective factors and violence outcomes.** With listwise deletion $N = 4,641$. CI, confidence interval; OR, odds ratio.
(DOCX)

**S4 Table. Sensitivity analysis of adjusted probabilities comparing single- and multiple-outcome modelling approach.**
(DOCX)

## Acknowledgments

We thank the adolescents and healthcare facilities who participated in this study, the field teams, the University of Oxford and University of Cape Town support teams, and Gaelen Pinnock, Angelique N. Chetty, and Leah de Jager for the graphic designs. Thank you to Catherine Maternowska, Sabine Rakotomalala, Elissa Miolene, and Andrew Hassett at the Global Partnership to End Violence for valuable inputs to the draft version, and to the World Food Programme for conceptual thinking.

## Author Contributions

**Conceptualization:** Lucie D. Cluver, Elona Toska, Franziska Meinck.

**Data curation:** Lucie D. Cluver, Elona Toska, Siyanai Zhou, Laurence Campeau.

**Formal analysis:** Lucie D. Cluver, William E. Rudgard, Yulia Shenderovich, Mark Orkin, Howard Taylor, Franziska Meinck.

**Funding acquisition:** Lucie D. Cluver.

**Methodology:** Lucie D. Cluver, William E. Rudgard, Yulia Shenderovich, Mark Orkin.

**Project administration:** Lucie D. Cluver, Franziska Meinck.

**Supervision:** Lucie D. Cluver.

**Writing – original draft:** Lucie D. Cluver, William E. Rudgard, Elona Toska, Siyanai Zhou, Laurence Campeau, Yulia Shenderovich, Mark Orkin, Chris Desmond, Alexander Butchart, Howard Taylor, Franziska Meinck, Lorraine Sherr.

**Writing – review & editing:** Lucie D. Cluver, William E. Rudgard, Elona Toska, Siyanai Zhou, Laurence Campeau, Yulia Shenderovich, Mark Orkin, Chris Desmond, Alexander Butchart, Howard Taylor, Franziska Meinck, Lorraine Sherr.

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
