## [Editor Report · Decision Letter 0]

23 Apr 2020

Dear Dr Cluver, 

Thank you for submitting your manuscript entitled "Violence prevention accelerators for children and adolescents in Africa" for consideration by PLOS Medicine.

Your manuscript has now been evaluated by the PLOS Medicine editorial staff and I am writing to let you know that we would like to send your submission out for external peer review.

Kind regards,

Artur Arikainen,

Associate Editor

PLOS Medicine

---

## [Decision Letter · Decision Letter 1]

25 Jun 2020

Dear Dr. Cluver,

Thank you very much for submitting your manuscript "Violence prevention accelerators for children and adolescents in Africa" (PMEDICINE-D-20-01470R1) for consideration at PLOS Medicine. 

[LINK]

In light of these reviews, I am afraid that we will not be able to accept the manuscript for publication in the journal in its current form, but we would like to consider a revised version that addresses the reviewers' and editors' comments. Obviously we cannot make any decision about publication until we have seen the revised manuscript and your response, and we plan to seek re-review by one or more of the reviewers. 

We expect to receive your revised manuscript by Jul 16 2020 11:59PM. Please email us (plosmedicine@plos.org) if you have any questions or concerns.

We look forward to receiving your revised manuscript. 

Sincerely,

Emma Veitch, PhD

PLOS Medicine

On behalf of Clare Stone, PhD, Acting Chief Editor,

PLOS Medicine

plosmedicine.org

*Please revise your title according to PLOS Medicine's style, this would normally include some indication of the study design (eg, here, an analysis based on 2 prospective cohorts) at the end of the title after a colon.

*Please structure your abstract using the PLOS Medicine headings (Background, Methods and Findings, Conclusions - "Methods and Findings" is a single subsection). In the last sentence of the Abstract Methods and Findings section, please include a brief comment about any key limitation(s) of the study's methodology.

*In the abstract, the acronym CI is used without having previously defined it (presumably, "confidence interval"). In addition, it would be good to explicitly state what the limits for the CIs are, presumably this is 95% but not always, sometimes 99% is seen.

*We'd suggest ensuring that the study is reported according to the STROBE guideline; in which case please include the completed STROBE checklist as Supporting Information. Please add the following statement, or similar, to the Methods: "This study is reported as per the Strengthening the Reporting of Observational Studies in Epidemiology (STROBE) guideline (SChecklist)." The STROBE guideline can be found here: http://www.equator-network.org/reporting-guidelines/strobe/. When completing the checklist, please use section and paragraph numbers, rather than page numbers.

*Please clarify if the analysis reported in this paper corresponded to that laid out in a prospectively set-out protocol or analysis plan? Please state this (either way) early in the Methods section.

Comments from the reviewers:

Reviewer #1: See attachment

Michael Dewey

Reviewer #2: Thank you for the opportunity to review the manuscript titled, "Violence prevention accelerators for children and adolescents in Africa" (PMEDICINE-D-20-01470R1). The manuscript reports on an innovative and compelling approach to informing the potential for policy and programmatic impacts on youth experiences of violence. The study has the potential to make significant contributions to the literature and address key questions about effective strategies to improve youth well-being and prevent violence. I recommend acceptance of the paper and note some points for the authors to consider in a revision.

1. Though WHO is the publisher of INSPIRE, WHO was not the sole institution to develop and release the technical package. The development was co-led by CDC and WHO, and all of the signatory agencies listed in INSPIRE co-released it, including UNICEF, PEPFAR, USAID, the World Bank, and others. This is one of the key hallmarks of INSPIRE, that it has been co-endorsed and co-released by multiple signatory agencies, allowing for consensus among key institutions in the most effective strategies to address violence against children (page 3).

2. Though the content of INSPIRE is primarily focused on prevention, there are also strategies (Response and support services) and programs directed at services for children who have experienced violence. I recommend revising the introduction to use language inclusive of both prevention and response, in places where only prevention of violence is noted (page 3).

3. I recommend including a bit more detail on the data collection procedures for the two studies. Were the questionnaires self-administered? What method of administration was used (paper and pencil, tablet, interview, etc.)? (Page 4)

4. From the description of the measures, it is not clear which are dichotomous/categorical and which are continuous. I checked the questionnaire on the Young Carers website, and given some of the measures use multiple items it is not possible to determine how the responses were coded and used. I recommend including more detail on how the specific outcomes and hypothesized violence prevention provisions were coded for the analyses (page 5).

5. The overall pattern of findings suggests that there are more similarities in "accelerator" effects across boys and girls than differences. I think this is a critical point that should be clearly noted in the discussion. However, I found this point to be less prominent. The first paragraph notes that parenting and food security were both accelerators. Based on the lack of significant differences in the gender interaction term it seems to be worth noting more clearly that the differences were in fact few (page 10).

Reviewer #3: I think that this complex analysis is very valuable to the field and worthy of publication. It is valuable to understand which protective factors are protective for multiple violence outcomes and how that can be leveraged to inform violence prevention programming and policy, particularly in low and middle income settings. I have some fairly minor questions/comments/suggestions for consideration below.

1. There are several references to "gender", when I think what is actually meant is biological sex in both the narrative (methods and results) and in the tables. I recommend changing "gender" to "sex".

2. The rates of sexual abuse and of transactional sex seem fairly low compared to other global surveys (e.g. VACS) and other literature from South Africa. This may be because of the young age of the sample, but it might be helpful to include the actual measures. For example, under sexual abuse in the methods section the sexual abuse measure is described as "two items from the Sexual Victimization module of the Juvenile Victimization Questionnaire". It would be helpful to include what those two measures are.

3. My largest comment is in the conceptualization of protective factors (self-reported on questionnaires) as being accelerators of INSPIRE. Based on my reading of the methods section, these study populations did not receive any violence interventions. Rather, they (general population youth) were asked about their experiences with violence victimization and their experience with various protective factors. These protective factors could have been resultant from interventions or not- we do not know because that was not assessed as far as I can gather. Therefore, I am struggling a little bit with the jump from showing in analysis that certain types of protective factors are protective for multiple types of violence to saying that these protective factors are therefore INSPIRE accelerators. In other words, I don't think that we can assume that because something was protective against violence outcomes means that it can therefore be conceptualized as an accelerator for prevention interventions. I actually re-read the methods section several times to understand if there was an intervention, but the way I read the methods section there was no intervention of positive parenting, supervision or food security- these were just self-reported. So I think there might need to be some additional explanation in the discussion section as to why the authors believe that evidence of existing protective factors (without known intervention) having an impact on violence outcomes in the general population can therefore be justification for assuming that interventions that target those protective factors will therefore have the same multiplicative effect on violence outcomes. I also found the term "provisions" to be a bit broad and recommend a different term such as "protective factors" or a similar term.

[LINK]

---

## [Decision Letter · Decision Letter 2]

18 Aug 2020

Dear Dr. Cluver,

Thank you very much for re-submitting your manuscript "Violence prevention accelerators for children and adolescents: A path analysis using two pooled cohorts in South Africa" (PMEDICINE-D-20-01470R2) for review by PLOS Medicine.

I have discussed the paper with my colleagues and the academic editor and it was also seen again by two reviewers. I am pleased to say that provided the remaining editorial and production issues are dealt with we are planning to accept the paper for publication in the journal.

[LINK]

We look forward to receiving the revised manuscript by Aug 25 2020 11:59PM. 

Sincerely,

Artur Arikainen, 

Associate Editor 

PLOS Medicine

plosmedicine.org

Requests from Editors:

1. Title: Please amend to: “Violence prevention accelerators for children and adolescents in South Africa: a path analysis using two pooled cohorts”

2. Abstract:

a. Please include the cohort recruitment dates.

b. Please include summary cohort demographics, eg. age, sex, n and % lost to follow-up.

c. Please quantify all results with p values.

d. Please replace “95% CIs” with “95% CI”

e. Line 71: "…predicted to significantly lower the prevalence…" should become "…was associated with a lower prevalence..." or similar

f. Please present the results for boys and sexual abuse prevalence, to match results presented for girls; mention whether or not significant.

g. In the last sentence of the Methods and Findings subsection, please include key limitation(s) of the study's methodology.

h. Conclusion: Please start this section with a summary of your findings, eg. “In this cohort study, we found that…”.

i. Avoid the use of causal language here and throughout the manuscript, eg. “may be effective” (line 87); use “may be associated with” or similar instead.

4. Methods:

a. Please give exact cohort recruitment dates, including day and month.

b. Please specify whether informed consent was written or oral.

c. Please provide more direct URL links to the participant questionnaires in all languages, or provide them as Supporting Information file(s).

d. Please provide more information on the pilot testing of questionnaires (line 209) in a Supporting Information file or cite published work, as appropriate.

e. Please state whether ethics approval was required or waived for this analysis.

5. When completing the STROBE checklist, please use section and paragraph numbers, rather than page numbers. Please upload the checklist as a separate file named ‘S1 Checklist’.

6. Results: Please quantify all results with p values.

7. Please avoid causal language, eg. “may protect against” (line 480); your study can only show association.

8. Some of the language is a little extreme (e.g., "groundbreaking in vision and impact" at line 574; "superb" at line 586) – please tone down as appropriate.

9. Please provide full access details (eg. DOI or URL) for references 3, 5, 6, 8, 12, 13, 24, 15, 18, 27, 33, 37, and 40. Re: references 27, 33 listed as forthcoming/in press, papers cannot be listed in the reference list until they have been accepted for publication or are otherwise publicly accessible (for example, in a preprint archive). The information may be cited in the text as a personal communication with the author if the author provides written permission to be named. Alternatively, please provide a different appropriate reference.

----

Comments from Reviewers:

Reviewer #1: The authors have addressed my points.

Michael Dewey

Reviewer #3: The manuscript is an important contribution to better understand how violence against children prevention programming can be maximized for the greatest impact. The author was very receptive to reviewers' feedback and made adjustments that addressed all of the feedback. I recommend publication of this important work.

[LINK]

---

## [Editor Report · Decision Letter 3]

18 Sep 2020

Dear Professor Cluver, 

On behalf of my colleagues and the academic editor, Dr. Greta Massetti, I am delighted to inform you that your manuscript entitled "Violence prevention accelerators for children and adolescents in South Africa: a path analysis using two pooled cohorts" (PMEDICINE-D-20-01470R3) has been accepted for publication in PLOS Medicine. 

PRODUCTION PROCESS

PRESS

PROFILE INFORMATION

Thank you again for submitting the manuscript to PLOS Medicine. We look forward to publishing it. 

Best wishes, 

Artur Arikainen, 

Associate Editor 

PLOS Medicine

plosmedicine.org